# Formulation of Polymers-Based Methotrexate Patches and Investigation of the Effect of Various Penetration Enhancers: In Vitro, Ex Vivo and In Vivo Characterization

**DOI:** 10.3390/polym14112211

**Published:** 2022-05-30

**Authors:** Muhammad Shahid Latif, Asif Nawaz, Sheikh Abdur Rashid, Muhammad Akhlaq, Asif Iqbal, Muhammad Jamil Khan, Muhammad Shuaib Khan, Vuanghao Lim, Mulham Alfatama

**Affiliations:** 1Advanced Drug Delivery Lab, Gomal Centre of Pharmaceutical Sciences, Faculty of Pharmacy, Gomal University, Dera Ismail Khan 29050, Pakistan; shahidlatif1710@gmail.com (M.S.L.); sheikhabdurrashid11@gmail.com (S.A.R.); dr.akhlaq@gu.edu.pk (M.A.); asifsaba188@gmail.com (A.I.); jamil@gu.edu.pk (M.J.K.); 2Faculty of Agriculture, Gomal University, Dera Ismail Khan 29050, Pakistan; 3Faculty of Veterinary and Animal Sciences, Gomal University, Dera Ismail Khan 29050, Pakistan; shuaibanatomy@gu.edu.pk; 4Advanced Medical and Dental Institute, Universiti Sains Malaysia, Kepala Batas 13200, Penang, Malaysia; 5Faculty of Pharmacy, Universiti Sultan Zainal Abidin, Besut Campus, Besut 22200, Terengganu, Malaysia

**Keywords:** hydroxypropyl methylcellulose (HPMC), Eudragit S-100 (ES-100), psoriasis, transdermal patches, methotrexate (MTX)

## Abstract

The present study aimed to prepare methotrexate-loaded transdermal patches with different blends of hydrophobic and hydrophilic polymers (Eudragit S-100 and hydroxypropyl methylcellulose) at different concentrations. The polymers employed in transdermal patches formulations served as controlled agent. Transdermal patches were prepared using the solvent casting technique. The suitable physicochemical properties were obtained from the formulation F5 (HPMC and Eudragit S-100 (5:1). Various penetration enhancers were employed in different concentrations to investigate their potential for enhancing the drug permeation profile from optimized formulations. A preformulation study was conducted to investigate drug–excipient compatibilities (ATR-FTIR) and the study showed greater compatibility between drug, polymers and excipients. The prepared patches containing different penetration enhancers at different concentrations were subjected for evaluating different physicochemical parameters and in vitro drug release studies. The obtained data were added to various kinetic models, then formulated patch formulations were investigated for ex vivo permeation studies, in vivo studies and skin drug retention studies. The prepared patches showed elastic, smooth and clear nature with good thickness, drug content, % moisture uptake and weight uniformity. The prepared transdermal patches showed % drug content ranging from 91.43 ± 2.90 to 98.37 ± 0.56, % swelling index from 36.98 ± 0.19 to 75.32 ± 1.21, folding endurance from 61 ± 3.14 to 78 ± 1.54 and tensile strength from 8.54 ± 0.18 to 12.87 ± 0.50. The formulation F5, containing a greater amount of hydrophilic polymers (HPMC), showed increased drug release and permeation and drug retention when compared to other formulated transdermal patch formulations (F1-F9). No significant change was observed during a stability study for a period of 60 days. The rabbit skin samples were subjected to ATR-FTIR studies, which revealed that polymers and penetration enhancers have affected skin proteins (ceramides and keratins). The pharmacokinetic profiling of optimized formulation (F5) as well as formulations with optimized concentrations of penetration enhancers revealed C_max_ ranged 167.80 ng/mL to 178.07 ± 2.75 ng/mL, T_max_ was 8 h to 10 h, and t^1/2^ was 15.9 ± 2.11 to 21.49 ± 1.16. From the in vivo studies, it was revealed that the formulation F5-OA-10% exhibited greater skin drug retention as compared to other formulations. These results depicted that prepared methotrexate transdermal patches containing different blends of hydrophobic and hydrophilic polymers along with different penetration enhancers could be safely used for the management of psoriasis. The formulated transdermal patches exhibited sustained release of drug with good permeations and retention profile. Hence, these formulated transdermal patches can effectively be used for the management of psoriasis.

## 1. Introduction

The administration of drug via skin has created great interest for both therapeutic effect and systemic drug delivery [1]. Transdermal delivery of drugs offers certain advantages including avoidance of drug degradation in the gastrointestinal tract (GIT), overcoming intra- and inter-subject variability, retention of the drug at the therapeutic concentration for a prolonged period, minimization of dosing frequency, enhanced compatibility, improved patient compliance, and termination of dose simply by removing the patch [2]. The formulated transdermal patches are kept over the surface of the skin to deliver sustained drug release and permeation to treat topical and systemic diseases [3]. In the formulation of transdermal patches, the polymers play an important role in sustaining, controlling and targeting drug deliveries and provides mechanical strength to the formulations [4]. Different blends of hydrophilic and hydrophobic polymers were employed for achieving controlled drug release behavior.

Hydroxypropyl methyl cellulose (HPMC) (K 100 M) is biodegradable, biocompatible and non-toxic derivative of cellulose ester. It is widely being used as an adhesive in the field of cosmetics formulations. HPMC possesses swelling, gelling and thickening properties that make it useful in sustained/controlled drug delivery. In the current study, high-grade HPMC (K100 M) was used to achieve controlled drug release from the prepared patches of methotrexate [5]. Eudragit S-100 is a non-biodegradable, non-absorbable, nontoxic and amorphous polymer. It possesses good taste and odor and offers sustained/controlled drug release. It is widely used in the preparation of enteric, gastrointestinal and transdermal drug delivery [6].

Various types of TDDS are currently available, including single-layer drug-in-adhesive, multi-layer drug-in-adhesive, reservoir, matrix and vapor patch. Single- and multi-layer drug-in-adhesive patches are the most commonly used, due to their simplicity and stability. In the single layer patches, the drug is integrated into the adhesive layer, which makes it accountable for the release of drugs and attachment to the skin, while in the multi-layer patches, there is an additional layer of a drug separated by a membrane. The ease of use and the absence of pain allow TDDS to be used in vulnerable patients, such as children and the elderly. However, the full potential of this delivery system is limited by the skin barrier [7].

The application of transdermal patches is an attractive way of drug administration for management of systemic diseases. The patches undergo natural or passive diffusion and allow the skin to penetrate the drug and enter into the systemic circulation [8]. Transdermal drug delivery also possesses some drawbacks. The higher molecular weight substances, i.e., >500 Dalton, fails to bypass stratum corneum. The transdermal patch fails to modify the stratum corneum nature and results in low drug plasma concentration [9].

The stratum corneum possesses barrier properties, and it prevents the entry of molecules from prepared transdermal formulations. In order to overcome the barrier nature of stratum corneum, various techniques such as iontophoresis, lipophilic lipid conjugates and liposomes have also been investigated for improving the capability of drug used topically [10].

For enhancing the skin, permeability penetration enhancers are used. Penetration enhancers cause changes in the barrier nature of stratum corneum and result in an enhanced flux. The penetration enhancers facilitate the diffusion of drugs and result in achieving a therapeutic concentration of drug in plasma [11]. In this present study, different penetration enhancers such as Oleic acid (OA), Eucalyptus oil (EO) and N-Methyl-2-pyrrolidone (NMP) were used as penetration enhancers.

Eucalyptus oil is a penetration enhancer that is soluble in oils, fats and dehydrated fats and insoluble in water and alcohol. Eucalyptus oil is used as a decongestant, expectorant and antiseptic. Eucalyptus oil causes disruption of intracellular lipids and results in fluidity of stratum corneum [12]. NMP is a chemical also penetration enhancer. NMP is used as penetration enhancers for enhancing drug plasma concentrations [13]. Oleic acid is a penetration enhancer that is widely being used in the preparation of transdermal formulations, because it causes disruption in the lipid structures of stratum corneum and results in achieving maximum drug permeation [14].

Methotrexate is used for various certain malignancies, autoimmune diseases such as rheumatoid arthritis and psoriasis. Oral application of methotrexate results in hepatic and gastrointestinal toxicities. However, systemic use of methotrexate results in severe side effects such as abdominal distress, nausea, ulcerative stomatitis, anemia, thrombocytopenia, leucopenia and depression disorders. Keeping in view all these demerits, it is desirable to administer methotrexate topically [15].

The transdermal patches containing different polymers and penetration enhancers at different concentrations were employed. In the formulation of transdermal patches, Polyethylene glycol (PEG-400) was used as plasticizer. The prepared patches were investigated for maximum drug release, permeation and retention. In psoriasis the deeper layer of skin was affected. The present study depicted that polymers and penetration enhancers employed exhibited outstanding and best patch formulation ability. Male albino rabbits were used for evaluating in vivo studies and skin drug retention. The prepared patches were investigated for various physicochemical parameters. The ATR-FTIR study was investigated for evaluation of skin interaction with drugs, polymers and penetration enhancers.

## 2. Materials and Methods

### 2.1. Materials

Methotrexate was obtained from Wilsons Pharmaceutical Pvt. Ltd., Pakistan. Ethanol, PEG-400 and Sodium-hydroxide (NaOH) (Dow Chemical Company., 693 Washington St #627, Midland, MI 48640, USA) and distilled water were used in the preparation of buffers and patches. Hydroxypropyl methyl cellulose (HPMC) (K-100 M) and Eudragit S-100 (E/S 100) (SIGMA-ALDRICH, INC., St. Louis, MO, USA, +1-314-771-5765, SIGMA-ALDRICH, Chemie Gmbh, Riedstr, Steinheim, Germany, +49-7329-970), served as controlled release agents. Eucalyptus oil (EO), N-methyl-2-pyrrolidone (NMP) and Oleic acid (OA) (SIGMA-ALDRICH, INC., Germany, +49-7329-970), were used as penetration enhancers.

### 2.2. Preparation of Methotrexate-Loaded Transdermal Patches

Solvent casting technique is a simple and easy method for preparation of transdermal patches. Methotrexate, polymers and excipients were weighed precisely using analytical balance (Shimadzu AX 200, Kyoto, Japan) (Table 1). The solvent system comprised equal amount of 10 mL of ethanol and distilled water. In the solvent system, weighed amount of polymers was dissolved and was stirred for 30 min using magnetic stirrer. Ph 7.4 was used for dissolution of drug. The drug aqueous phase was added with the mixture of excipients, polymers and solvents. PEG-400 was added as plasticizer for prevention of brittleness of formulated transdermal patches. In the sonicator (D-78224 Singen Germany, +49-7731-8820), solution was placed for removal of entrapped air bubbles. In the Petri dish, the clear solution was poured carefully. In oven, these Petri dishes were kept at 40 °C temperature for a time period of 6 h. The final prepared patches were kept in desiccator until further use. The formulation F5 (HPMC: Eudragit S-100; 5:1) was our optimized formulation which showed higher values for % drug content, tensile strength, % swelling index and folding endurance. Similarly, patch formulations with optimized patch (F5) were prepared with the addition of different penetration enhancers (Oleic acid, N-methyl-2-pyrrolidone and Eucalyptus oil at different concentrations, respectively.

### 2.3. Preformulation Study (Compatibility Studies)

ATR-FTIR study was performed for investigation of incompatibilities between drug polymers and excipients used in the preparation of methotrexate-loaded transdermal patches (F1–F9). The ATR-FTIR a spectrum of pure drug (methotrexate) was compared with its physical mixture. Each spectrum was evaluated with total 32 scan at resolutions ranging between 4000 and 600 cm^−1^.

### 2.4. Physicochemical Evaluation of Transdermal Patches of Methotrexate with and without Penetration Enhancers

The following physicochemical analysis was carried out for the formulated transdermal patches:

#### 2.4.1. Physical Appearance

All the methotrexate-loaded transdermal patch formulations were inspected physically.

#### 2.4.2. Weight Variations

Analytical weighing balance (Shimadzu AX 200, Kyoto, Japan, +81-75-823-1111) was used for the investigation of weight variations in between formulated patches. The data obtained was averaged for confirming weight variation values [16].

#### 2.4.3. Thickness

Vernier caliper (Kevelaer, Germany, +49-2832-92390) was used for the evaluation of patches. Uniformity of patch thickness was ensured by measuring patches thickness at different six places. The result obtained was calculated and averaged (mean ± SD) [16].

#### 2.4.4. Folding Endurance

This test was performed over all prepared patches. This test is performed to ensure plasticizer efficacy in transdermal patch formulations. The folding of formulated transdermal patches at same place were carried out until breaking and cracking in the patch formulations appears, and the number of folding of patch showed value of folding endurance. Following equation best demonstrate folding endurance [16].
F = d (1)
where F is folding endurance and d is number of folds required to break or crack the patch.

#### 2.4.5. Surface pH

The pH meter (InoLab^®^ Bremen, Germany) was utilized for evaluating surface pH. The pH determination for the prepared transdermal patches is of utmost importance because an alkaline and acidic pH might contribute towards skin irritation. Test tubes were used for evaluating surface pH. An amount of 2 mL of distilled water and a 2 cm^2^ portion of transdermal patch were kept in the test tubes. After 2 h, filtration was carried out for removal of excess water. The obtained results were calculated and averaged (mean ± SD) [16].

#### 2.4.6. Swelling Index Evaluation

The individual weights of prepared transdermal patches were taken, then patches were placed in desiccator and aluminum chloride was also placed in desiccator for three days. The formulated patches were removed carefully from the desiccator. The initial and final weight of formulated patches exhibited the swelling index value for formulated patches. Following equation was used for calculating % swelling index [17]:*SI* = (*w1* − *w2*)/*w2* × *100*
(2)
where *w1* is indicating patch initial weight. *w2* is indicating final weight. The result obtained was calculated and averaged (mean ± SD).

#### 2.4.7. Tensile Strength

The tensile strength was evaluated for formulated patches using pulley system. The formulated patch was measured using scales. The formulated patch was tied to the hooks while another side of the patch was tied to the rope, the rope was attached to the weighing pan. The weight is slowly increased in the weighing pan until creak or break appears in the formulated patches. The value of tensile strength was calculated from the weight of pan. The thread pointer method was used for evaluation of percent elongation. Tensile strength value was determined using following equation [17]:*Tensile Strength* = *F*/[*a* × *b* (*1* + *L*/*I*)] (3)
where *F* is patch breaking forces, *a* is patches width and *b* is patches thickness (cm). *L* is patches length (cm), and *I* is patches elongation (cm) before occurrence of cracking or breaking. Percent elongation value was calculated by following equation:% *Elongation* = (*L*
*f* − *Li*)/*Li* × *100*
(4)
where *Li* is patches initial lengths and *Lf* is patch final length.

#### 2.4.8. Water Vapor Transmission Rate

Oven-dried transmission cells were carried out in this study. The formulated patches were individually weighed. Potassium chloride was kept in desiccator, and 1 g of anhydrous calcium chloride was fixed at brim. The transmission cells were kept in the desiccator. Transmission cells were removed from the desiccator after specified interval of time, i.e., 6, 12, 24, 36, 48, and 72 h. The transmission cells were reweighed at the end of the study [17].

#### 2.4.9. Percentage Moisture Content

Initially formulated patches were weighed separately. The formulated patches were kept in desiccator. Silica was also placed in desiccator for 24 h. The patches were weighed individually for evaluating constant weight of prepared patches. The % content moisture value was estimated from the difference in the weight of prepared transdermal patches. This equation was used to investigate the value of % moisture content [17].
% *Moisture Content* = (*wi* − *wf*)/*wi* × *100*
(5)
where *wi* is representing patches initial weights, and *wf* is representing patches final weights.

#### 2.4.10. Estimation of Drug Content

Estimation of drug content is of utmost importance in terms of long-term stability of the products. Phosphate buffer Ph 7.4 was used for the estimation of drug content of formulated transdermal patches. In volumetric flask, drug and phosphate buffer 7.4 was taken and placed in sonicator for a time period of 6 h, then solution was filtered using filter paper. The filtrate was analyzed at 303 nm wavelength using UV-visible spectrophotometer (Shimadzu AX 200, Kyoto, Japan) [18].

### 2.5. Stability Study

The stability of formulated patches was carried out for 60 days. This study was conducted at different temperatures and humidity. The relative humidity was created in the stability chamber by placing Petri dish in chamber along with cotton soaked with water. The hygrometer was fitted in such a way that its knob was dipped in the Petri dish and display was kept outside to investigate the humidity value. Due to temperature variations, water evaporates and hygrometer displays the humidity value. In case of 75 ± 5% RH value, more water was added. When RH value exceeds 80%, then Petri dish was removed from stability chamber. After 20-day interval, the physical appearance and drug content were evaluated. The data obtained were calculated and averaged (mean ± SD, *n* = 3). The procedure for the estimation of drug content is already being discussed in in Section 2.4.10.

### 2.6. Skin Irritation Studies

The study was performed with proper NOC approval from Gomal Centre of Pharmaceutical Science, Faculty of Pharmacy, Gomal University, Dera Ismail Khan, KP, Pakistan. For this study, healthy male albino rabbits weighing 2–2.5 kg were used. Standard food was given to the rabbits [18]. The temperature was maintained at 25 ± 2 °C, with relative humidity controlled at 50 ± 10% for all rabbits. The test animals were divided into five groups. The study was completed in a week. Visual scale was used for grading of skin irritation. Skin irritation was calculated visually as follows: no skin irritation, “0”, slight skin irritations, “1”, well defined skin irritations, “2”, moderate skin irritations, “3” and scar formation was indicated with “4” [18].

### 2.7. In Vitro Drug Release Study

Franz Diffusion Cell Apparatus (Perme-Gear, Hellertown, PA, USA) was used for investigation of in vitro drug release study. Artificial membrane (Tuffryn membrane) (diameter 2.5 mm and pore size 0.45 µm) was employed on the diffusion cells and 2.5 cm^2^ area of formulated transdermal patch was fixed over artificial membrane. The receptor compartments were filled with phosphate buffer (pH 5.5). The receptor compartments were maintained at 32 ± 0.5 °C temperature with continues stirring of magnetic stirrer beads were fixed over 100 rpm. Then, 2 mL aliquots were obtained at specified time intervals, i.e., 0.5, 1, 1.5, 2, 4, 8, 12, 16, and 24 h, respectively. The sink condition of receptor compartments was carried out with the addition of fresh buffer (pH 5.5). The collected samples were evaluated using UV visible spectrophotometer at 303 nm wave length [18].

### 2.8. Kinetics of Drug Release

The drug release mechanisms of formulated transdermal patches of methotrexate were evaluated with the use of following kinetic models.

#### 2.8.1. Zero Order Kinetics

Zero order kinetic equation or model may be used for the constant release rate characterization or representation of a dosage form that do not disintegrate. Those dosage forms which follows zero order kinetic, release the constant amount of drug per unit time. To achieve prolonged pharmacological action in sustained release dosage forms, zero order model is an ideal kinetic model.

Zero order kinetics was carried out using following equation:*W* = *k1* × *t*
(6)
where *W* is representing releasing of drug, *k1* is representing as constant of zero order kinetics and t is drug releasing time [19,20].

#### 2.8.2. First Order Kinetics

The first order kinetics equations or model was first proposed by Gibaldi and Feldman (1967) and later by Wagner (1969), where this model was used for description and characterization of absorption and release or elimination of certain drugs from biological systems. This was used for the investigation of absorption and elimination rate. This kinetic model is used in sink conditions. First order kinetics was carried out using following equation:*ln* (*100* − *W*) = *ln 100* − *k2t*
(7)
where *W* is representing drug releasing, *k2* is representing constant of First order kinetics and t is drug releasing time [21,22].

#### 2.8.3. Hixon–Crowell Model

To evaluate the drug release with changes in the surface area and the diameter of the particles/tablets, Hixon–Crowell (1931) recognized that the particle regular area is proportional to the cubic root of its volume, designing an equation as:(*100* − *W*) *1*/*3* = *100* (*1*/*3*) − *k3t*
(8)
where *W* is representing drug releasing time, *k3t* representing relationships of Hixon–Crowell and surface volume, and *t* is representing drug releasing time [23].

#### 2.8.4. Higuchi Model

Higuchi used this model for the 1st time. The Higuchi or Diffusion model describes the dissolution of drugs in suspension from non-eroding matrix such as ointment base. This method is also applicable for description of dissolution rates of other pharmaceutical dosage forms other than ointment and also for description of dissolution process mechanism of modified drug delivery systems. Higuchi model was carried out using following equation:*W* = *k4t* × *1*/*2*
(9)
where *W* is representing drug release, *k4* is representing constant of Higuchi rate of dissolutions, and *t* is representing drug release [24].

#### 2.8.5. Power Law Equation

Korsmeyer, 1983, and Rigter and Peppas, 1987, used this equation. This model is also known as Korsmeyer–Peppas equation. It is simple semi-empirical model that creates a relationship between drug release and elapsed time with an exponential function. In this model, the value of *n* showed amount of drug released from formulated patches. The value of *n* equal to 0.45 showed Fickian diffusion. When the release data is of *n* is less than 0.89, then the release mechanism is known to be Non-Fickian, or anomalous, diffusion; when data obtained from release data for *n is* equal to 0.89, then the release mechanism is known to be Case II transport; when the release data obtained for *n* is greater than 0.89, then the release mechanism is known as Super Case II transport.

Power law was carried out using following equation:*Mt*/*M*∞ = *k4 t n*
(10)
where *Mt/M∞* is representing drug release, *k4* is representing constant for power, and *n* is representing diffusion exponent. The value of *n* depicts transport behavior of drugs [25,26].

### 2.9. Ex Vivo Permeation Study

Rabbit Skin Preparations

The study was performed with proper NOC approval from Gomal Centre of Pharmaceutical Science, Faculty of Pharmacy, Gomal University, Dera Ismail Khan, KP, Pakistan. For this study, healthy male albino rabbits weighing 2–2.5 kg were used. Standard food was given to the rabbits [18]. The test animals were maintained at room temperature with relative humidity. The rabbits were anesthetized with the injection of overdose of ketamine and xylazine and the skin was excised surgically. In warm water, the excised rabbit skin was placed for removal of adhered fats. The excised skin was washed with 0.9% sodium chloride solution and was kept at −20 ± 1 °C [16].

The investigation of ex vivo permeation of methotrexate from formulated transdermal patches was carried out using Franz Diffusion Cell Apparatus. In the receptor compartment, phosphate buffer (pH 7.5) was filled. The temperature of receptor compartments was maintained at a temperature of 37 ± 0.5 °C, and stirring of magnetic stirrer beads were fixed over 100 rpm. The excised skin was placed over receptor cells and 2.5 cm^2^ of prepared methotrexate transdermal patch was placed on excised rabbit skin. Then, 2 mL aliquots were taken in test tube at specified time intervals, i.e., 0.5, 1, 1.5, 2, 4, 8, 12, 16, and 24 h, respectively. In order to maintain sink condition, fresh buffer (pH 5.5) was replaced. The collected samples were evaluated using UV visible spectrophotometer at 303 nm wave length [16].

### 2.10. Drug Retention Study

This study was carried out for all the formulated transdermal patch formulations. The skin obtained from ex vivo study was dried at 50 °C for 30 s, and with the help of scalpel, the scraping of skin was performed for separating epidermis and dermis. The epidermis and dermis were kept in glass vials separately and methanol was added for extraction of drug. The collected aliquots were evaluated using UV visible spectrophotometer at 303 nm wavelength, and the results obtained were calculated and averaged (mean ± SD) [27].

### 2.11. Skin ATR-FTIR Spectroscopy

ATR-FTIR spectra study was carried out for the formulated transdermal patches in the frequencies ranges from 4000 to 650 cm^−1^ with spectral resolutions of 4 cm^−1^. The software used for assigning the peaks was Perkin Elmer Spectrum Version 6.0.2 (Perkin Elmer, Waltham, MA, USA). the skin after completion of ex vivo permeation studies was collected from Franz diffusion cell apparatus, and then it was washed with solution of phosphate buffer (pH 7.4). The skin was further blotted with soft tissue and was kept onto internal reflection element of ATR-FTIR spectrophotometer (Perkin Elmer, Waltham, MA, USA). Normalization of skin was carried out for minimizing inter-samples variations and the force gauge of 80 N was applied to all formulations.

### 2.12. In Vivo Studies

The study was performed with proper NOC approval from Gomal Centre of Pharmaceutical Science, Faculty of Pharmacy, Gomal University, Dera Ismail Khan, KP, Pakistan. For this study, healthy male albino rabbits weighing 2–2.5 kg were used. Standard food was given to the rabbits [18]. The test animals were maintained at room temperature with relative humidity. The rabbits were anesthetized with the injection of overdose of ketamine and xylazine and back region of rabbits were shaved using electrical trimmer.

The rabbit used in the in vivo studies were characterized into two groups; each group consists of 06 rabbits. Group-A was applied with the optimized formulation (F5 Control). The Group-B was applied with the optimized patch formulation with penetration enhancers (experimental group). From rabbits, 0.5 mL blood was obtained at pre-determined time interval and was centrifuged for collection of plasma. In the plasma, methanol was added and vortexed for 20 min. The vortexed plasma was centrifuged for 3 min at 5000 rpm. The estimation of drug plasma content was carried out on HPLC by adding supernatant to the filtrations process, dissolved in 0.5 mL HPLC mobile phase pH 6 (0.2 M Na_2_HPO_4_ and 0.2 M citric acid; 2:1), and acetonitrile in 90:10 *v*/*v*. Overdose of ketamine and xylazine injection (i.m.) were administrated to the rabbits. The skin obtained was washed with 0.9% sodium chloride solution and was cut into small pieces. The skin was dipped in distilled water for 24 h and was filtered. The residual drug concentrations were determination using HPLC [16].

### 2.13. Statistical Analysis

In this study, triplicate results were obtained, calculated and averaged (mean ± SD). SPSS version 16 software (IBM, Chicago, IL, USA) was used for statistical analysis. *p* < 0.05 was considered as significant. The statistical tool used in the study was one-way ANOVA/post hoc analysis using Tukey’s honestly significant difference test. For the evaluation of in vivo studies, Two-way ANOVA was used.

## 3. Results and Discussion

The present study focuses on methotrexate transdermal patch formulations by employing both hydrophilic and hydrophobic polymers used in various concentrations to obtain the optimized control release formulation. The optimized formulation (F5) was further treated with various penetration enhancers employed in different concentrations to evaluate their potential for enhanced skin penetration as well as drug retention in skin deeper layers. The penetration enhancers employed in different concentrations for optimized formulation (F5) included (Eucalyptus Oil (E-Oil), N-methyl-2-pyrrolidone (NMP) and Oleic acid (OA), respectively.

### 3.1. Drug—Excipients Compatibility Studies


Pure drug methotrexate and its physical mixture with different polymers were carried out for ATR-FTIR analysis to identify any sort of incompatibilities. In this study, different physical mixtures of hydrophilic and hydrophobic polymers were employed in the preparation of transdermal patches. The characteristics peaks of methotrexate were showed at 3450 cm^−1^ (O–H stretching of carboxyl group) (Figure 1). At 3080 cm^−1^, primary amine, N-H starching group, was showed. –C=O carboxylic group stretching and C=O amide group stretching was showed in the range of 1670–1600 cm^−1^. The amidic band N-H band is showed in the range 1550–1500 cm^−1^ and partial overlapping with C=C stretching was observed. The carboxylic group –C-O stretching was showed in the range of 1400–1200 cm^−1^. The formulated patches showed good molecular structure and confirmed purity of the drug (methotrexate). The drug and polymers peaks were kept preserved; hence, ATR-FTIR study showed no incompatibility between drug and excipients. Hence, the study showed no sort of incompatibilities in between drug (methotrexate) and polymers (Eudragit S-100 and HPMC) used in the preparation of transdermal patches.

### 3.2. Physicochemical Evaluation of Transdermal Patches without Penetration Enhancers

The prepared patches were carried out for evaluating various physicochemical parameters (Table 2). The physical appearance of all the formulated patches gave satisfactory result. The prepared patches were flexible, homogenous, opaque, non-sticky and smooth natured. The skin pH was found in the range 5.1 to 5.9. The surface pH is of utmost importance for transdermal formulation to avoid skin irritations. The formulated patches thickness is a crucial influencing factor on drugs uniformity. The thickness of transdermal patches has influence on drug release, permeation, retention and drug diffusivity via skin. The thickness of formulated patches ranged between 0.47 ± 0.09 mm and 0.71 ± 0.05 mm. The patch thickness consistency is obtained by the low standard deviation measurements [28]. The weight variations of the formulated patches depicted from 70.68 ± 0.05 mg to 95.44 ± 1.10 mg. The presence of increased concentration of HPMC results in increased patch weight. This might be due to the reason that HPMC is hydrophilic. The hydrophilic polymers causes water retention and results in increased weight of patches, while Eudragit S-100 is hydrophobic in nature and increase in concentration of Eudragit S-100 formulate thin patches matrixes [29]. The % moisture content in the formulated patches containing maximum concentrations of HPMC polymer was found higher, the reason might be HPMC is hydrophilic in nature and results in greater amount of moisture absorption as compared to hydrophobic polymers (Eudragit S-100). The prepared patches exhibited moisture content values from 9.01 ± 0.19% to 12.45 ± 1.83% [30]. The prepared patches showed % swelling index from 36.98 ± 0.19% to 75.32 ± 1.21%. The study showed that increase in concentration of hydrophilic polymers results in increased swelling index. Eudragit S-100 results in less moisture uptake and prevents the formulated patches from bulkiness and microbial contaminations [31]. The formulated patches exhibited drug contents in the range of 92.57 ± 3.22% to 98.37 ± 0.56%. All the formulated patches exhibited uniform drug content with in pharmacopoeial limits. This study depicted that formulated transdermal patches are capable of offering drug content uniformity, hence convenient for transdermal application. The formulated transdermal patches showed water vapor transmission rate from 2.13 ± 0.15 g/m^2^/24 h to 4.82 ± 0.61 g/m^2^/24 h. The study depicted that formulation F5 showed increased rate of water vapor transmission. This might be due to the presence of maximum amount of hydrophilic polymer (HPMC). The formulation F9 exhibited low rate of water vapor transmission, due to presence of maximum concentration of hydrophobic polymer [32]. The formulated patches depicted good folding endurance value. The results indicated that the formulated patches would not break or crack with skin folding. The greater folding endurance was obtained in those formulated patches containing higher amount of hydrophilic polymers (HPMC) as compared to the formulations containing higher amount of hydrophobic polymer (Eudragit S-100) [33]. Folding endurance of formulated patches showed 61 ± 3.14 to 74 ± 2.32 (Table 2). Tensile strength is of utmost importance in providing mechanical strength to the formulated transdermal patches. The addition of plasticizer to the formulated patches offers strength to the formulated patches. The result depicted that higher amount of Eudragit S-100 results in less tensile strength [34]. Those formulated patches containing higher amount of hydrophilic polymers exhibited higher value of tensile strength. Our formulated transdermal patches exhibited tensile strength from 8.54 ± 0.18 kg/cm^2^ to 12.23 ± 0.23 kg/cm^2^.

### 3.3. Stability Study Profile of Methotrexate-Loaded Patches (F1–F9)

The prepared patches were investigated for physical changes and drug contents (Table 3). Various temperature and humidity levels were used to complete stability studies for formulated transdermal methotrexate-loaded patches for a time period of 60 days. The study depicted that the prepared patches remained flexible, elastic natured and showed almost the same drug content value at the start and at the end of this study. The study revealed that the formulated patches are physically and chemically stable and all the patches showed insignificant change in drug content (*p* < 0.05).

### 3.4. Skin Irritation Study

All the prepared transdermal patch formulations were carried out for skin irritancy potential. The prepared patches showed no skin irritancy as compared to the standard irritant formalin (Figure 2). The formalin produces severe erythema and edema. The study resulted that theses formulated patches (F1–F9) were nonirritant to the skin and were suitable candidates for use in transdermal applications [35].

### 3.5. In Vitro Drug Release Study

This study predicts the release behavior of drug from the prepared system. The pattern and rate of drug release is also calculated from this study. In this study, the % drug release was plotted against time. Initially, burst release form the formulated patches was observed; the reason might be due to the quick dissolution of surface drug. This initial burst release of drug is beneficial in case of transdermal applications. The constant drug release from a formulation results in zero order kinetics. Fick’s law is used to predict the amount of drug dissolved.

The formulation F5 (HPMC: Eudragit; 5:1) showed maximum drug release (79.96%) within 24 h (Figure 3). The reason might be due to the presence of maximum amount of hydrophilic polymer (HPMC). The formulation F9 (HPMC: Eudragit 1:5) showed minimum drug release (54.43%) within 24 h. The reason might be attributed to the addition of maximum amount of hydrophobic polymer. Hence, F5 showed maximum amount of drug followed by sustained drug release from the formulated patch. This difference in the drug release behavior of all the formulated patches might be due to the presence of cross linkages networks of polymeric chains. Different polymeric blends used in the formulation of transdermal patches have varied diffusion pathways that results in varying the release and diffusion duration. From transdermal system the molecular diffusion via polymeric matrixes is a simple, effective and reliable mean for achieving sustained or controlled drug release. This study depicted that the formulation F5 (HPMC: Eudragit S-100; 5:1) showed maximum drug release followed by controlled drug release for 24 h.

### 3.6. Drug Release Kinetics

The obtained data was subjected into the different kinetic models to investigate the release pattern of drug (Table 4). The data were fitted to different kinetic models. Linear equation was showed by all formulated patches. Power law equation was better suited to our data. Anomalous, non-Fickian release mechanism was observed for MTX patches. The value of ‘n’ obtained from formulated transdermal patches ranged between 0.738 and 0.838. The study depicted that formulated transdermal patches exhibited erosion and swelling mechanism.

### 3.7. Ex Vivo Permeation Study

The formulated patches were subjected for ex vivo permeation studies (Figure 4). Among all the formulated patches, the formulation F5 (HPMC: Eudragit S-100; 5:1) exhibited greater amount of drug permeation from patch formulation as compared to formulation F1 (HPMC: Eudragit S-100; 1:1). The enhanced drug release is attributed to the existence of greater amount of hydrophilic polymer and formulation F1 comprises the same amount of hydrophilic and hydrophobic polymers. HPMC produces a gel layer by capturing the water molecules and results in the maximum amount of drug escaping from the system. The formulation F1 served as control formulation and exhibited 22.35% cumulative amount of drug permeation, while formulation F5 exhibited 34.25% cumulative amount of drug permeation and exhibited a statistically significant difference (*p* < 0.05). The formulation F1 exhibited a flux value of 11.54 µg/h/cm^2^, which is less than the required flux of 20.11 µg/h/cm^2^. The study depicted that the increase in the concentration of hydrophilic polymers results in an increased flux value. The formulation F5 contained the maximum amount of hydrophilic polymer (HPMC) and exhibited a two-fold higher flux as compared to the targeted flux. The presence of hydrophobic polymer (Eudragit S-100) results in decreased values for flux. The formulation F9 (HPMC: Eudragit S-100; 1:5) results in 20.43% of drug permeation compared to F1 formulation (22.35%). The value of flux obtained in case of F9 was exhibited to be 9.23 µg/h/cm^2^. In the case of formulation F5, the higher amount of methotrexate permeation is due to the presence of the highest amount of hydrophilic polymer (HPMC) as it causes fluidization of skin and results in altering the barrier nature of stratum corneum, while in case of formulation F9, the least amount of drug permeation is due to the presence of a higher amount of hydrophobic polymer (Eudragit S-100).

### 3.8. Drug Retention Studies

The formulated patches (F1-F9) were carried out to investigate the retention of methotrexate in the dermal and epidermal layers of the skin (Figure 5). The study depicted that change in the concentration of polymers resulted in varying the retention of methotrexate in the dermal and epidermal layers of the skin. The formulation F5 (HPMC: Eudragit; 5:1) exhibited maximum amount of the drug retention (*p* ≤ 0.05) as compared to the formulation F1(HPMC: Eudragit S-100; 1:1) and formulation F9 (HPMC: Eudragit S-100; 1:5), respectively. The reason might be due to the presence of the maximum amount of hydrophilic polymer (HPMC) causing fluidization of skin and resulting in enhanced drug retention.

### 3.9. Skin ATR-FTIR Studies

The bands of lipids matrixes were evaluated with the ATR-FTIR studies. The lipid and proteins bands of stratum corneum are exhibited at different wave numbers. The chain of lipid hydrocarbons, symmetrical and asymmetrical CH_2_ stretching vibration showed at 2918.05 cm^−1^ and 2855.08 cm^−1^ wave numbers. Similarly, the amide I (C=O) stretching vibrations and amide II C-N stretching and N-H bending in proteins of stratum corneum were showed at 1650 cm^−1^ and 1550 cm^−1^ wave numbers. The fluidization of stratum corneum was showed by the shifting of C_2_ stretching vibrations to the elevated wave numbers. The same result can be obtained in case of formulation F1, F5 and F9, as showed in (Figure 6i). The result depicted significant shifting of peaks in case of formulation F5; this is due to having the maximum amount of hydrophilic polymer, which is attributed to enhanced drug release, permeation and retention.

In case of formulations F1, F5 and F9, the ATR-FTIR study of dermis also exhibited that fluidization has been occurred at wave number 3300–3380 cm^−1^, representing OH/NH, and 2850–2930 cm^−1^, representing asymmetrical CH_2_ skin lipids, keratin, and ceramide, as showed in Figure 6ii. The study also depicted that application of formulated transdermal patches have significantly affected the epidermis and dermis layer of the skin.

### 3.10. Preparation of MTX Patches with Penetration Enhancers

Among all the prepared formulation, the formulation F5 was regarded as an optimized formulation based on the results of various patch formulations designed with polymeric blends of hydrophilic as well as hydrophobic polymers (HPMC and Eudragit S-100). For further physicochemical assessment as well as in vivo studies, the optimized formulation F5 was treated with various penetration enhancers. Following this, penetration, enhancers were incorporated in the preparation of transdermal patches (Eucalyptus oil, N-methyl-2-Pyrrolidone and Oleic acid), respectively. They were added to patches in different concentrations, as shown in Table 5.

### 3.11. Physicochemical Evaluation of MTX Patches with Penetration Enhancers

The prepared transdermal patches with penetration enhancers were carried out for the following physicochemical evaluation (Table 6). The prepared patches showed non-sticky, transparent, smooth, homogenous, flexible and elastic nature. The pH of the formulated transdermal patches ranged between 5.1 and 5.7. The surface pH is of utmost importance for transdermal formulation to avoid skin irritations. The thickness of prepared patches was from 0.68 ± 0.04 to 0.72 ± 0.07 mm. The thickness of formulated patches depicted uniformity of thickness. It was depicted that the addition of penetration enhancers did not affect the appearance and thickness of the formulated patches. The formulated transdermal patches showed uniformity of weight. The addition of penetration enhancer to the formulation of transdermal patches slightly increases the weight of the patches. The weight of the formulated patches ranged between 92.37 ± 0.08 and 97.93 ± 0.07.

The moisture content in the formulated patches varied slightly. The slightly increase in moisture content was found with the addition of penetration enhancers. The moisture content values for the formulated patches ranged between 11.18 ± 1.54% to 12.77 ± 1.12% [30]. The formulated patches showed the swelling index in the range between 69.29 ± 0.76 and 77.65 ± 0.32. The rate of swelling index affects the drug release from the prepared patches. Low swelling index rates were obtained in case of all formulated patches, this beneficial for long-term storage of the patches. A slight increase was found in the swelling index value, this might be due to the presence of penetration enhancers in the formulated patches. The uniform drug distribution in the formulated transdermal patches results in achieving sustained or controlled drug delivery. The drug content values of prepared patches were from 95.66 ± 2.43% to 98.87 ± 2.21%. The prepared patches exhibited uniformity of drug content with negligible patch variability.

The formulated transdermal patches showed water vapor transmission rates ranging from 2.46 ± 0.43 g/m^2^/24 h to 4.76 ± 0.41 g/m^2^/24 h. The study depicted that addition of penetration enhancers have insignificant difference over rate of water transmission in the formulated transdermal patches. All the formulated showed folding endurance ranged in between 69 ± 1.22 to 78 ± 1.54. The formulated patches pass the folding endurance test. The study depicted that transdermal patches can be applied to the skin and patch would not crack nor break during in vivo studies. The formulated patches showed good tensile strength ranged in between 10.23 ± 0.16 kg/cm^2^ to 12.87 ± 0.50 kg/cm^2^. The efficacy of plasticizer is also depicted from a good value of tensile strength. The addition of PEG-400 to the formulated patches results in a reduction in patch brittleness. The study depicted that the addition of 30% *w*/*w* of plasticizer (PEG-400) produces flexible, smooth and elastic transdermal patches.

### 3.12. Skin Irritation Study of Transdermal Patches of Methotrexate with Penetration Enhancers

All the prepared transdermal patch formulations were carried out for skin irritancy potential. The prepared patches showed no skin irritancy as compared to the standard irritant formalin (Figure 7). The formalin produces severe erythema and edema. The study resulted that the formulated patches (F1–F9) were non-irritant to the skin and were suitable candidates for use in transdermal applications. Moreover, the additions of various penetration enhancers (Eucalyptus oil, N-methyl-2-pyrrolidone and Oleic acid) in various concentrations have no contribution towards erythema or edema. Draize patch test was used for evaluating skin irritation potential of the prepared patches. In the case of all patches, the values obtained were below 2, which indicates that prepared patches were safe to be used tropically.

### 3.13. In Vitro Drug Release of Methotrexate-Loaded Patches Containing Penetration Enhancers

The formulated transdermal patches were carried out for investigating rate and extent of drug (methotrexate) release behavior. The different concentrations of different penetration enhancers employed in the preparation patches were evaluated for methotrexate release from the formulated patches. The study depicted that addition of penetration enhancers have insignificant difference over the release of the methotrexate from formulated patches, as compared to formulation F5 (Control).

However, an increase in average release (%) along with an increase in the concentration of penetration enhancers were found, as showed in Figure 8a–c. The formulation F5 (control) exhibited 61.23% drug release in 24 h. The formulation containing different penetrations at different concentrations exhibited insignificant difference in drug release (*p* > 0.05). However, a slight increase was found with various concentrations of penetration enhancers. The 5% concentration of Eucalyptus oil exhibited 68.99% drug release, 5% N-methyl-2-pyrrilidone concentration exhibited 65.98% and 10% Oleic acid exhibited 69.77 drug releases.

### 3.14. Ex Vivo Permeation Study of MTX Patches Containing Penetration Enhancers

The use of penetration enhancers in topical application is widely being used to achieve higher permeation and retention of drug. The major barrier in transdermal drug delivery is the stratum corneum. The chemical penetration enhancers are usually employed in the preparation of tropical formulation to breech the stratum corneum barrier nature. The chemical penetration enhancers are inactive in nature; they augment the drug permeation by changing proteins and lipids of the stratum corneum. Numerous approaches have been used for breaching the skin barrier. The widely used method is the incorporation of penetration enhancers. In this study, the effect of Eucalyptus oil, NMP and Oleic acid was incorporated at various concentrations. The Eucalyptus oil was employed in the transdermal formulation at different concentrations such as 1%, 3%, 5% and 10% *w*/*w*, respectively.

This study depicted that the rate of drug permeation increases from 1% to 5% of the penetration enhancer, and after 5%, insignificant drug permeation was exhibited when the concentration of Eucalyptus oil was increased up to 10%, as showed in Figure 9a. The flux value of drug (methotrexate) increased 9.77 times and showed significant difference as compared to the control formulation F5, which contained no penetration enhancers (*p* < 0.05). Penetration enhancers have a significant effect over enhancement ratios. The formulation (F5-E Oil 5%) was found to be 9.77 (rabbit skin). The study depicted that addition of Eucalyptus oil increases the methotrexate permeation as compared to control formulation F5 (HPMC: Eudragit S-100; 5:1). The enhanced skin permeation is attributed to the combined effect of more concentration of hydrophilic polymer and penetration enhancers. The most promising result was obtained when the Eucalyptus oil was used in 5% *w*/*w*. Previous studies also reported that the enhancement effect of Eucalyptus oil is concentration dependent. Terpenes cause enhanced skin diffusion, since cineol is the active constituents of Eucalyptus oil. The cineol has a low boiling point, and it interacts with the lipid component of the stratum corneum and results in week cohesiveness and self-association [36].

The study depicted that 5% of N-methyl-2-pyrrilidone exhibited maximum drug (methotrexate) permeation, as showed in Figure 9b. The NMP exhibited a 7.05-fold increase in permeation, as compared to the controlled formulation F5 (HPMC: Eudragit S-100; 5:1). It was observed that at a low concentration, NMP showed insignificant difference in the permeation value, as compared to the control formulation F5. The use of 5% NMP as a penetration enhancer exhibited a significant difference as compared to controlled formulation (p < 0.05) [37]. The enhancing effect of NMP might be due to its improved portioning characteristics. This process might be facilitated by the formation of hydrogen bonding between drug and NMP [38]. The enhancement in the drug permeation might be due to the lipoidal fluidization of stratum corneum.

The study depicted that 10% of Oleic acid exhibited maximum drug permeation as compared to control formulation F5 (HPMC: Eudragit; 5:1) at 24 h, as showed in Figure 9c. The Oleic acid exhibited 8.5 times greater flux as compared to the control formulation F5 (*p* < 0.05). The study depicted that an increase in the concentration of Oleic acid is concentration dependent. The increase in the higher concentration of Oleic acid results in a higher amount of drug permeation. The transdermal drug delivery of lumiracoxib as a chemical enhancer has been evaluated and showed that 10% of Oleic acid showed maximum drug permeation for BCS Class-IV drugs. The result was in close agreement with our results [39].

### 3.15. Drug Retention Analysis of MTX Patches Containing Penetration Enhancers

The formulated patches were carried out for skin drug retention studies (Figure 10). Skin drug retention was carried out for both dermis and epidermis layers of skin. The study depicted that a higher amount of methotrexate was retained in the deeper layer of the skin as compared to the other formulations (one-way ANOVA, *p* < 0.05). The flux of drug is directly influenced by the epidermis and dermis layer of the skin. These layers of skin acts as a reservoir for drug passage to the systemic circulations [39].

In the case of psoriasis, methotrexate has an inhibitory effect at the site of inflammation. Mostly, psoriasis is found in the epidermal and dermal layer of the skin. The maximum amount of methotrexate retention in these two layers of the skin exhibited promising effects to treat psoriasis. The study depicted that a transdermal formulated patch containing 5% Eucalyptus oil, 5% NMP and 10% Oleic acid showed the highest amount of methotrexate retention in the deeper layers of the skin. Hence, it is concluded from the study that 5% of Eucalyptus oil, 5% of NMP and 10% of Oleic acid have the potential to breech the barrier nature of skin.

### 3.16. Effect of MTX-Loaded Patch Formulations on Skin Structure

The formulated patches were incorporated for ATR-FTIR studies. The protein and lipid molecular vibrations are showed at different wave numbers. The CH_2_ symmetrical and asymmetrical stretching vibrations bands of lipid hydrocarbons were showed at 2918.05 cm^−1^ and 2855.08 cm^−1^. Furthermore, the amide I (C=O) stretching vibrations and amide II (C-N stretching and N-H bending) in the stratum corneum proteins were showed at 1650 cm^−1^ and 1550 cm^−1^. The formulated transdermal patches results in fluidization of stratum corneum. The fluidization occurred in the lipoidal layers of stratum corneum. The CH_2_ stretching vibrations were shifted to the higher wave number that depicted the fluidization of the stratum corneum, as showed in Figure 11i. The ATR-FTIR result of the optimized formulation exhibited that the major peak of the epidermis and the dermis were shifted to the higher wave number. This shifting of the peaks exhibited that fluidization of the stratum corneum occurred. The formulated patches affected the barrier nature of the stratum corneum by altering protein and lipoidal layer of stratum corneum. This fluidization results in ease passage of methotrexate via stratum corneum and greater amount of drug retention can be achieved.

The study showed that fluidization of the skin occurred in the case of dermis. The fluidization occurred at 3380 cm^−1^ representing OH/NH functional groups. The fluidization of ceramides, keratin and lipids takes place at 2850–2930 cm^−1^. The study depicted that application of all patch formulations (Formulations F5 control, F5-E-Oil-5%, F5-NMP-5%, and F5-OA-10%) have affected the skin stratum corneum and have facilitated permeation of drug from the formulated transdermal patch formulations as showed in Figure 11ii [40].

### 3.17. In Vivo Studies

Among all the formulations, three patch formulations were selected as optimized patch formulations. The optimization of patches was based on different physicochemical parameters, drug release, permeation and retention studies. The three optimized formulations were carried out for in vivo studies. The comparative bioavailability study was carried out from serum data of all the prepared patch formulations. The formulated patches were investigated for different pharmacokinetic parameters C_max_, t_max_, AUC, t_1/2_, kel and MRT (Table 7). The time profile and mean plasma concentration of all the formulated patches are showed in Table 7, and on the basis of MRT, AUC_0-t_ and biological half-life values, the formulated patches were ranked as F5-OA-10% > F5-E-Oil 5% > F5-NMP-5%.

The control formulation F5 showed 167.80 ± 3.41 ng/mL plasma levels, while optimized patch formulations showed plasma levels of methotrexate ranging from 171.59 ± 3.57 ng/mL to 178.07 ± 2.75 ng/mL (Table 7). The enhanced methotrexate permeation is useful in the identification of systemic circulations. Transdermal application of methotrexate is beneficial in the case of psoriasis, as more drug is retained in the deeper layer of the skin and the disease is treated locally. The study depicted low plasma residual concentration from the optimized formulated patches and results in minimizing systemic side effects. The biological half-life observed in the case of formulation F5 (Control) was 4–10 h^−1^, while an increase in the value of biological half-life was observed from formulated transdermal patches and ranged in between 15.91 ± 2.11 h^−1^ to 21.49 ± 1.16 h^−1^. This result indicates that the formulated transdermal patches will stay longer and will produce controlled effects. The formulation F5 (Control) exhibited the least elimination rate constant of 0.040 ± 0.02 h^−1^, while formulated patches exhibited increased elimination rate constants and ranged from 17.15 ± 2.51 h^−1^ to 21.49 ± 1.16 h^−1^.

The formulation F5 (Control) exhibited the least amount of mean residence time, i.e., 11.65 ± 0.52 h, while the transdermal patch formulations exhibited mean residence time ranging between 22.21 ± 0.61 h and 25.63 ± 0.51 h. The extended mean residence time results in controlled and extended drug activities. The greater AUC values depicted enhanced bioavailability of medication, and this might be due to the avoidance of stomach degradation and bypassing the hepatic 1st pass effect. The results obtained from this study were in accordance to the studies reported earlier.

Two-way ANOVA was used as a significant difference (*p* < 0.01) was observed in between the test products, but not within the tested products, when AUC0-12, AUC0-24, AUC0-∞, t1/2, kel and MRT data were obtained from the serum and were taken into consideration. When Cmax was taken into consideration, no significant difference was observed (*p* > 0.01) (Table 7).

## 4. Conclusions

This study revealed that transdermal patch formulations can best be suited for the management of chronic skin aliments such as psoriasis. In this disorder, the delivery of medication is uncertain, owing to rigid growth of the stratum corneum. So, we concluded in this study that adequate use of polymeric blends as well as optimum concentration of penetration enhancers could overcome the short comings of transdermal drug deliveries for potentially managing psoriasis. Methotrexate was chosen for its efficacy as well as evaluation of its better penetration and augmented dermatokinetics. Topical patch formulations containing methotrexate by employing different concentration of HPMC and Eudragit S-100 polymers as well as optimum concentrations of various penetration enhancers were studied. All batches of transdermal patch with and without penetration enhancers showed good physicochemical properties. The study showed that in vitro drug release is directly proportional to the use of polymer types, their concentration and their combination ratios. The optimized formulations and its further extension with various penetration enhancers showed enhanced drug release, permeation as well as skin drug retention mainly by effecting skin proteins (ceramides and proteins). Thus, the patch formulation could be a suitable choice to treat psoriasis. The topical administration of optimized patch formulation demonstrated reduced plasma concentration, leading to better patient compliance by reducing the likelihood of systemic toxic investigation.

## Figures and Tables

**Figure 1 polymers-14-02211-f001:**
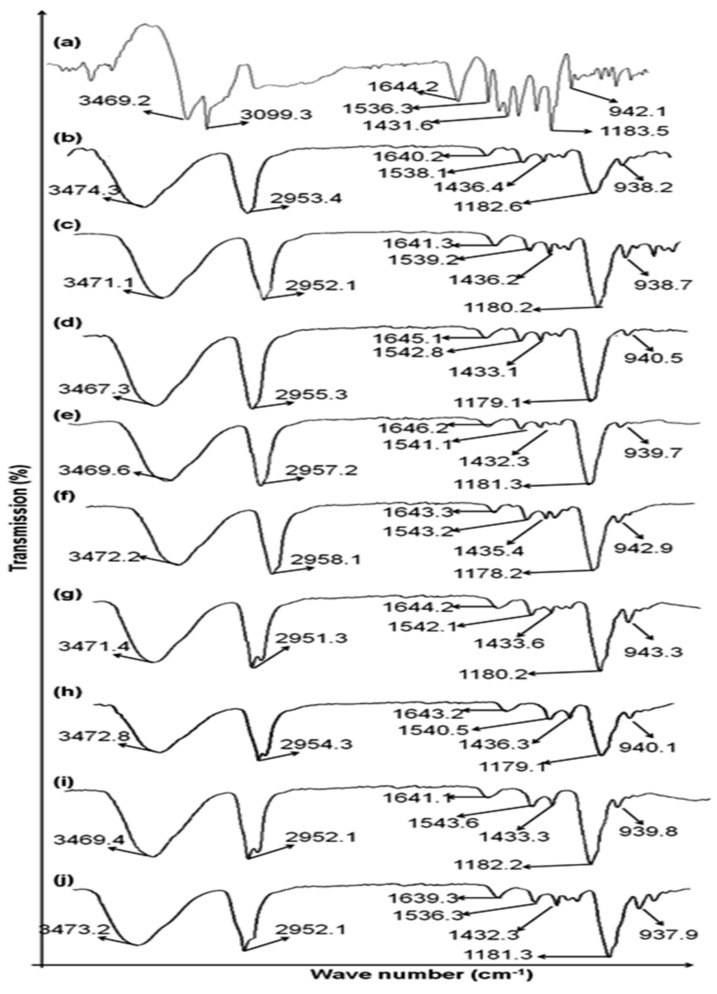
ATR–FTIR spectrum (**a**) Methotrexate (**b**) F1 (**c**) F2 (**d**) F3 (**e**) F4 (**f**) F5 (**g**) F6 (**h**) F7 (**i**) F8 (**j**) F9.

**Figure 2 polymers-14-02211-f002:**
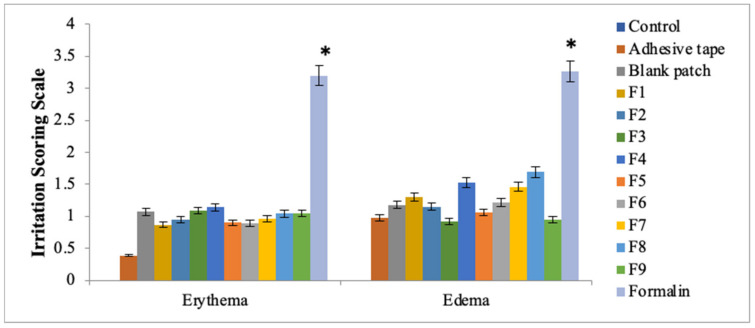
Skin irritation studies: erythema and edema. Data are expressed as mean ± SD; *n* = 3. One-way ANOVA followed by post hoc Tukey test; *p* < 0.05). (* *p* < 0.05).

**Figure 3 polymers-14-02211-f003:**
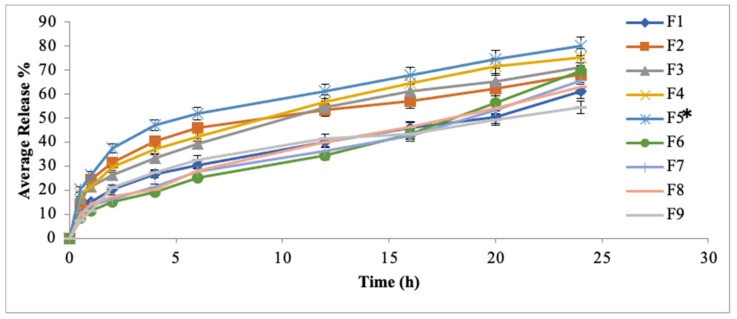
In vitro drug release of MTX patches. Data are expressed as mean ± SD; *n* = 3. One-way ANOVA followed by post hoc Tukey test (*p* < 0.05), F5 vs. F1. (* *p* < 0.05).

**Figure 4 polymers-14-02211-f004:**
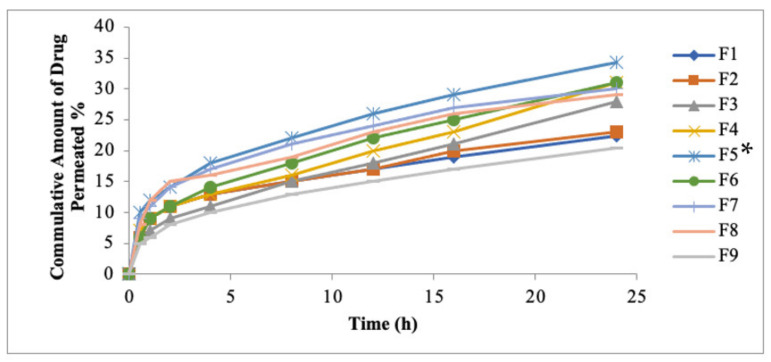
Ex vivo permeation of MTX patches. Data are expressed as mean ± SD; *n* = 3. One-way ANOVA followed by post hoc Tukey test (*p* < 0.05), F5 vs. F1. (* *p* < 0.05).

**Figure 5 polymers-14-02211-f005:**
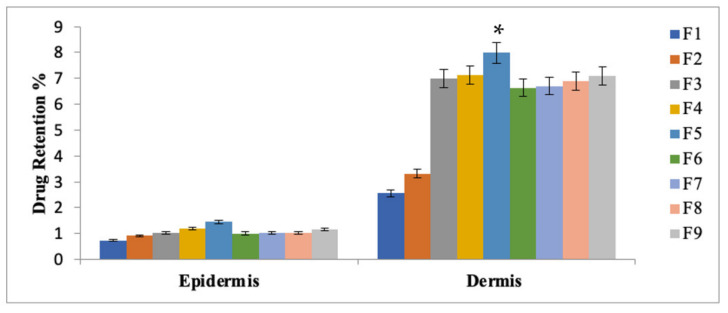
Skin drug retention of MTX patches. Data are expressed as mean ± SD; *n* = 3, one-way ANOVA followed by post hoc Tukey test (*p* < 0.05), F5 vs. F1. (* *p* < 0.05).

**Figure 6 polymers-14-02211-f006:**
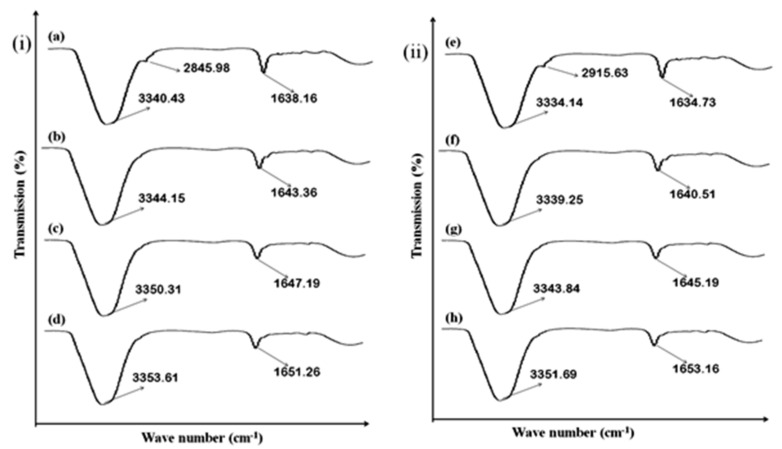
(**i**)**.** Rabbit skin epidermis ATR-FTIR: (**a**) untreated, (**b**) treated with F1, (**c**) treated with F5, (**d**) treated with F9, (**ii**)**.** Representation of rabbit skin dermis FTIR: (**e**) untreated, (**f**) treated with F1, (**g**) treated with F5, (**h**) treated with F9, data are expressed as mean ± SD; *n* = 3.

**Figure 7 polymers-14-02211-f007:**
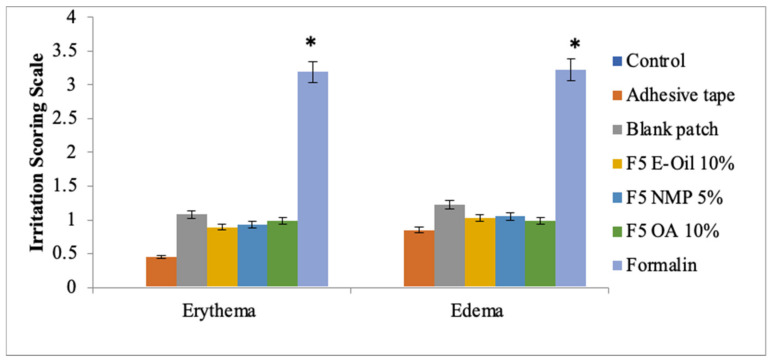
Skin irritation studies: erythema and edema. Data are expressed as mean ± SD; *n* = 3. One-way ANOVA followed by post hoc Tukey test; *p* < 0.05. (* *p*<0.05).

**Figure 8 polymers-14-02211-f008:**
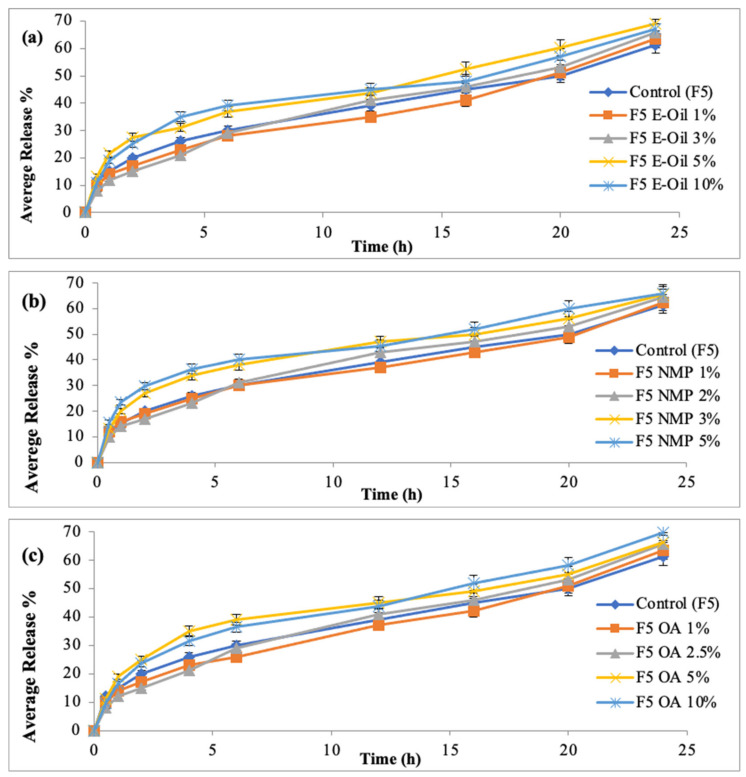
Release profiles of methotrexate from formulated patches using (**a**) EO, (**b**) NMP and (**c**) OA as penetration enhancers. Data are expressed as mean ± SD; *n* = 3. One-way ANOVA followed by post hoc Tukey test (*p* > 0.05).

**Figure 9 polymers-14-02211-f009:**
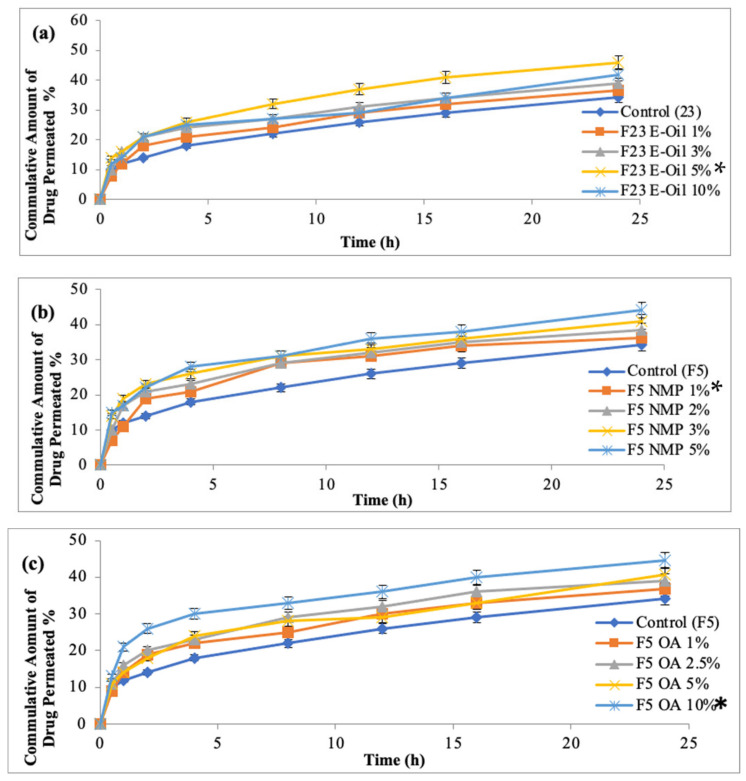
Permeation profiles of methotrexate from formulated patches using (**a**) E-Oil, (**b**) NMP and (**c**) OA as penetration enhancer. Data are expressed as mean ± SD; *n* = 3. One-way ANOVA followed by post hoc Tukey test (*p* < 0.05). (* *p*<0.05).

**Figure 10 polymers-14-02211-f010:**
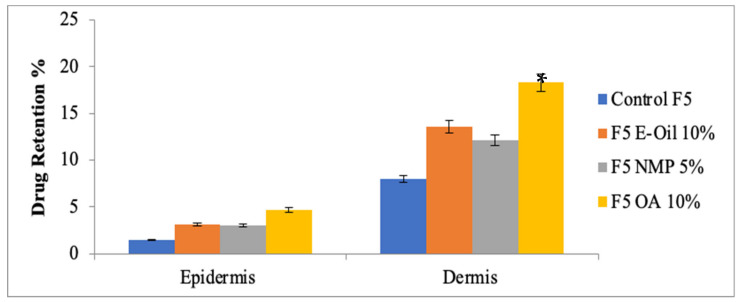
Skin drug retention analysis of optimized MTX patches with optimized concentrations of penetration enhancer. One-way ANOVA followed by post hoc Tukey test (*p* < 0.05). Data are expressed as mean ± SD; *n* = 3. (* *p*<0.05).

**Figure 11 polymers-14-02211-f011:**
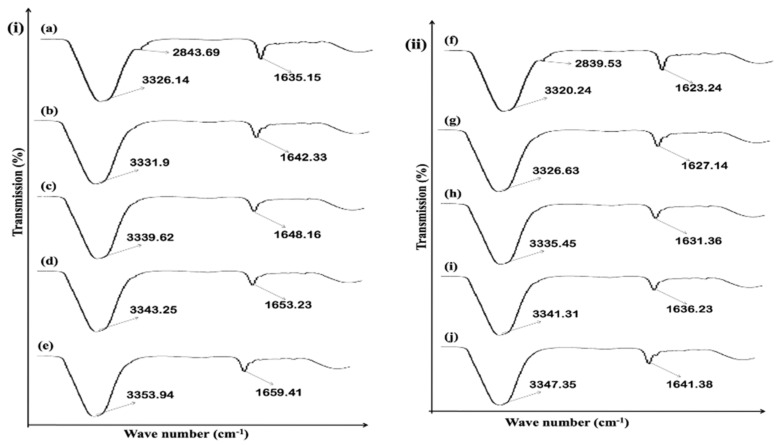
(**i**): Representation of rabbit skin epidermis FTIR: (**a**) untreated (**b**) treated with F5 (Control), (**c**) treated with F5-EO-5%, (**d**) treated with F5-NMP-5%, and (e) treated with F5-OA-10%; (**ii**): representation of rabbit skin dermis FTIR: (**f**) untreated, (**g**) treated with F5 (Control), (**h**) treated with F5-EO-5%, (**i**) treated with F5-NMP-5%, and (**j**) treated with F5-OA-10%. Data are expressed as mean ± SD, *n* = 3.

**Table 1 polymers-14-02211-t001:** Compositions of formulated transdermal patches.

		Polymers			Plasticizer	Solvents (*v*/*v*) mL	
F Code	MTX (mg)	HPMC (mg)	ES-100 (mg)	Combination HPMC/ES-100	PEG-400 (%)	Ethanol	Distilled Water
F1	5	100	100	1:1	30	10	10
F2	5	100	200	1:2	30	10	10
F3	5	100	300	1:3	30	10	10
F4	5	100	400	1:4	30	10	10
F5	5	500	100	1:5	30	10	10
F6	5	100	200	2:1	30	10	10
F7	5	100	300	3:1	30	10	10
F8	5	100	400	4:1	30	10	10
F9	5	100	500	5:1	30	10	10

**Table 2 polymers-14-02211-t002:** Characterization of methotrexate patches.

Characteristics									
F. Code	pH	Thickness (mm)	Weight Uniformity (mg)	% Moisture Content	% Swelling Index	% Drug Content	Water Transmission Rate g/m^2^/24 h	Folding Endurance(d)	Tensile Strength Kg/cm^2^
F1	5.5	0.57 ± 0.12	80.14 ± 0.13	10.67 ± 1.42	61.85 ± 1.46	94.58 ± 3.14	2.13 ± 0.15	66 ± 3.14	11.23 ± 0.23
F2	5.7	0.61 ± 0.07	85.61 ± 0.05	11.12 ± 1.73	63.58 ± 0.29	96.23 ± 2.08	3.88 ± 0.34	69 ± 1.21	11.54 ± 0.18
F3	5.6	0.64 ± 0.02	89.54 ± 0.12	11.53 ± 1.67	65.61 ± 1.26	97.81 ± 1.17	4.23 ± 0.67	71 ± 2.14	11.93 ± 0.56
F4	5.7	0.67 ± 0.03	91.23 ± 0.07	11.92 ± 1.64	69.82 ± 0.37	98.37 ± 0.56	4.61 ± 0.52	73 ± 2.31	12.13 ± 0.93
F5	5.8	0.71 ± 0.05	95.44 ± 0.10	12.45 ± 1.83	75.32 ± 1.21	98.54 ± 1.31	4.82 ± 0.61	74 ± 3.16	12.64 ± 0.28
F6	5.6	0.54 ± 0.03	77.75 ± 0.09	10.38 ± 1.10	56.16 ± 0.85	97.32 ± 1.33	2.61 ± 0.17	67 ± 2.32	9.55 ± 0.39
F7	5.3	0.52 ± 0.10	76.34 ± 0.06	10.13 ± 1.48	46.23 ± 0.33	96.58 ± 2.36	2.43 ± 0.32	65 ± 2.59	8.96 ± 0.46
F8	5.4	0.50 ± 0.05	72.43 ± 0.08	9.78 ± 1.73	41.63 ± 1.59	94.81 ± 1.46	2.28 ± 0.27	62 ± 1.43	8.59 ± 0.72
F9	5.2	0.47 ± 0.09	70.68 ± 0.05	9.01 ± 1.47	36.98 ± 0.19	92.57 ± 3.22	2.15 ± 0.63	61 ± 1.23	8.21 ± 0.33

Data are expressed as mean ± SD.

**Table 3 polymers-14-02211-t003:** Stability studies of methotrexate patches.

F. Code	Initial % Drug Content	25 ± 2 °C (60 ± 5%RH)			40 ± 2 °C (75 ± 5%RH)		
		15 Days	30 Days	60 Days	15 Days	30 Days	60 Days
F1	94.58 ± 3.14	94.36 ± 2.34	94.15 ± 2.32	93.86 ± 2.23	94.43 ± 3.23	94.26 ± 2.23	93.72 ± 2.54
F2	96.23 ± 2.08	96.14 ± 1.34	95.83 ± 1.34	95.64 ± 3.54	95.93 ± 2.43	95.74 ± 3.34	95.52 ± 3.56
F3	97.81 ± 1.17	97.66 ± 2.43	97.43 ± 3.54	97.25 ± 2.23	97.64 ± 3.45	92.35 ± 2.43	91.22 ± 2.34
F4	98.37 ± 0.56	98.21 ± 3.45	98.10 ± 2.45	97.96 ± 1.65	98.16 ± 4.23	97.83 ± 4.54	97.62 ± 3.54
F5	98.54 ± 1.31	98.37 ± 2.45	98.25 ± 3.34	98.03 ± 4.23	98.34 ± 1.65	98.16 ± 2.23	97.94 ± 2.23
F6	97.32 ± 1.33	97.15 ± 1.35	96.85 ± 1.65	96.66 ± 2.45	97.13 ± 2.23	96.74 ± 3.65	96.63 ± 2.65
F7	96.58 ± 2.36	96.23 ± 2.34	96.16 ± 2.76	95.93 ± 4.34	96.33 ± 3.54	96.10 ± 1.34	95.85 ± 4.45
F8	94.81 ± 1.46	94.64 ± 2.24	94.48 ± 3.34	94.25 ± 2.23	94.65 ± 4.76	94.45 ± 2.76	94.16 ± 2.65
F9	92.57 ± 3.22	92.36 ± 2.34	92.14 ± 1.76	91.93 ± 2.65	92.23 ± 2.23	92.12 ± 3.34	91.84 ± 2.34

Data are expressed as mean ± SD.

**Table 4 polymers-14-02211-t004:** Kinetic profiling of methotrexate patches.

F. Code	Zero order		1st Order		Higuchi		Hixon–Crowell		Korsmeyer–Peppas		
	K ± SD	R^2^	K ± SD	R^2^	K ± SD	R^2^	K ± SD	R^2^	K ± SD	R^2^	*n*
F1	1.37 ± 0.012	0.968	1.28 ± 0.116	0.541	1.38 ± 0.142	0.981	1.45 ± 0.252	0.991	0.12 ± 0.321	0.985	0.798
F2	1.25 ± 0.116	0.972	1.24 ± 0.169	0.613	2.54 ± 0.227	0.985	1.32 ± 0.143	0.984	0.163 ± 0.012	0.997	0.838
F3	1.11 ± 0.154	0.989	1.18 ± 0.125	0.638	2.52 ± 0.481	0.988	1.28 ± 0.268	0.989	0.034 ± 0.158	0.991	0.764
F4	1.23 ± 0.128	0.978	1.26 ± 0.116	0.584	2.52 ± 1.214	0.991	1.34 ± 0.132	0.993	0.043 ± 0.321	0.993	0.784
F5	1.26 ± 0.148	0.983	1.84 ± 0.143	0.564	1.45 ± 0.029	0.99	1.64 ± 0.154	0.985	0.15 ± 0.148	0.995	0.784
F6	1.14 ± 0.141	0.964	1.42 ± 0.136	0.663	2.53 ± 0.124	0.976	1.37 ± 0.182	0.984	0.167 ± 0.124	0.991	0.782
F7	1.25 ± 0.213	0.981	1.31 ± 0.142	0.689	2.16 ± 0.136	0.984	1.41 ± 0.268	0.983	0.035 ± 0.132	0.994	0.804
F8	1.32 ± 0.186	0.984	1.26 ± 0.158	0.586	2.86 ± 0.006	0.991	1.46 ± 0.321	0.988	0.081 ± 0.163	0.991	0.738
F9	1.15 ± 0.124	0.986	1.15 ± 0.186	0.594	2.22 ± 0.124	0.989	1.32 ± 0.143	0.992	0.141 ± 0.183	0.994	0.752

Data are expressed as mean ± SD; *n* = 3.

**Table 5 polymers-14-02211-t005:** Preparation of methotrexate transdermal patch formulations with penetration enhancers.

		Polymers		Penetration Enhancers (*w*/*v*)			Plasticizer	Solvents (*v*/*v*)	
F Code	MTX (mg)	HPMC (mg)	ES-100 (mg)	EO	NMP	OA	PEG-400 (%)	Ethanol	Distal Water
F5 EO 1%	5	500	100	1			25	20	20
F5 EO 3%	5	500	100	3			25	20	20
F5 EO 5%	5	500	100	5			25	20	20
F5 EO 10%	5	500	100	10			25	20	20
F5 NMP 1%	5	500	100		1		25	20	20
F5 NMP 2%	5	500	100		2		25	20	20
F5 NMP 3%	5	500	100		3		25	20	20
F5 NMP 5%	5	500	100		5		25	20	20
F5 OA 1%	5	500	100			1	25	20	20
F5 OA 2.5%	5	500	100			2.5	25	20	20
F5 OA 5%	5	500	100			5	25	20	20
F5 OA 10%	5	500	100			10	25	2	20

**Note:** Methotrexate (MTX), Hydroxypropyl-methylcellulose (HPMC), Eudragit S-100 (ES-100), Eucalyptus oil (EO), N-methyl-2-pyrrolidone (NMP) and Oleic acid (OA), Polyethylene glycol-400 (PEG-400).

**Table 6 polymers-14-02211-t006:** Characterization of methotrexate patches with penetration enhancers.

Characteristics									
F. Code	pH	Thickness (mm)	Weight Uniformity (mg)	% Moisture Content	% Swelling Index	% Drug Content	Water Transmission Rate	Folding Endurance	Tensile Strength Kg/cm^2^
F5 E-Oil 1%	5.3	0.70 ± 0.03	94.45 ± 0.01	12.77 ± 1.12	72.65 ± 0.12	98.34 ± 1.33	4.34 ± 0.43	78 ± 1.54	11.15 ± 0.11
F5 E-Oil 3%	5.2	0.72 ± 0.02	93.76 ± 0.05	11.67 ± 1.32	74.47 ± 0.32	97.24 ± 2.23	3.54 ± 0.35	76 ± 1.43	10.23 ± 0.16
F5 E-Oil 5%	5.3	0.71 ± 0.04	94.45 ± 0.03	12.65 ± 1.22	75.43 ± 0.21	95.67 ± 2.20	4.35 ± 0.32	78 ± 1.34	11.27 ± 0.64
F5 E-Oil 10%	5.1	0.69 ± 0.09	96.43 ± 0.07	11.45 ± 1.54	69.22 ± 0.65	98.54 ± 1.76	3.88 ± 0.76	71 ± 1.56	12.87 ± 0.50
F5 NMP 1%	5.5	0.72 ± 0.05	95.22 ± 0.06	12.41 ± 1.87	73.21 ± 0.38	97.76 ± 2.43	4.76 ± 0.41	69 ± 1.22	12.76 ± 0.87
F5 NMP 2%	5.6	0.69 ± 0.06	94.28 ± 0.08	11.62 ± 1.98	74.47 ± 0.37	96.69 ± 1.76	3.61 ± 0.68	70 ± 1.67	10.57 ± 0.79
F5 NMP 3%	5.7	0.68 ± 0.04	93.69 ± 0.09	12.21 ± 1.56	72.87 ± 0.84	96.60 ± 1.65	2.98 ± 0.66	74 ± 1.89	11.55 ± 0.74
F5 NMP 5%	5.3	0.72 ± 0.07	96.56 ± 0.04	11.18 ± 1.54	77.65 ± 0.32	95.43 ± 2.90	4.54 ± 0.53	72 ± 1.75	10.89 ± 0.44
F5 OA 1%	5.2	0.70 ± 0.07	94.19 ± 0.06	10.98 ± 1.33	75.32 ± 0.39	98.87 ± 2.21	3.54 ± 0.78	78 ± 1.13	10.29 ± 0.24
F5 OA 2.5%	5.6	0.71 ± 0.03	97.93 ± 0.07	12.43 ± 1.27	69.29 ± 0.76	96.67 ± 1.76	2.46 ± 0.43	74 ± 1.47	11.38 ± 0.56
F5 OA 5%	5.4	0.71 ± 0.02	92.37 ± 0.08	11.27 ± 1.54	72.76 ± 0.43	95.66 ± 2.43	3.43 ± 0.77	71 ± 1.53	10.83 ± 0.59
F5 OA 10%	5.6	0.69 ± 0.01	96.89 ± 0.02	12.35 ± 1.32	71.54 ± 0.27	97.54 ± 1.32	3.37 ± 0.49	73 ± 1.32	12.64 ± 0.43

Data are expressed as mean ± SD; *n* = 3.

**Table 7 polymers-14-02211-t007:** Evaluation of pharmacokinetic parameters of control and optimized methotrexate patches.

Formulations	T_max_(h)	C_max_(ng/mL)	t_1/2_	K (h^−1^)	AUC_0-t_ (ng/mL.h)	MRT (h)
Control (F5)	10	167.80 ± 3.41	15.91 ± 2.11	0.040 ± 0.02	2765.15 ± 132.1	11.65 ± 0.52
F5 E-Oil 5%	8	176.13 ± 4.51	17.15 ± 2.51	0.031 ± 0.01	2884.51 ± 131.5	23.81 ± 0.61
F5 NMP 5%	8	171.61 ± 2.15	18.93 ± 2.13	0.032 ± 0.02	2693.14 ± 147.3	22.21 ± 0.61
F5 OA 10%	8	178.07 ± 2.75	21.49 ± 1.16	0.027 ± 0.05	2949.27 ± 192.7	25.63 ± 0.51
Two-way ANNOVA ‘P’		ns	<0.05 (s)	<0.05 (s)	<0.01 (HS)	<0.01 (HS)

**Note:** C_m_ MRT: mean residence time. AUC_0-t_: area under the plasma concentration-time plot from 0 h to time. K(h^−1^): elimination rate constant. t_1/2_: elimination half-life. C_max_: peak plasma drug concentration. T_max_: time at which C_max_ was observed.

## Data Availability

Not applicable.

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
