# Peer review of "Formulation of Polymers-Based Methotrexate Patches and Investigation of the Effect of Various Penetration Enhancers: In Vitro, Ex Vivo and In Vivo Characterization"

_polymers, 2022, doi:10.3390/polym14112211_

Round 1

Reviewer 1 Report

The authors have presented transdermal patches of methotrexate for potential treatment of psoriasis. The work presented is novel and add progress to the science. The paper can be accepted after minor revision. Following points can be considered while revising the manuscript;

  1. In material section, please mention the grade of HPMC.
  2. What was the rational of using the specific grade of HPMC?
  3. In skin irritation study, please mention the relative humidity.
  4. For in vitro drug release study, please mention properties of Tuffryn membrane, such as thickness of membrane, pore size or cut-off MW etc.
  5. Why Power Law model for spherical shape was considered for transdermal patch? The 'n' value should corresponds to flat surface (film) and the subsequent discussion should be based on it.  

Author Response

Reply to the comments of Reviewer 1.

I acknowledge your efforts regarding sparing your precious time to evaluate our research article and gave valuable assessment to make it publishable. Your comments are commendable and helped us a lot to refine this manuscript as well as will help us in future to keep in mind all these parameters to make the research articles acceptable in each and every aspect. The entire manuscript has been thoroughly checked and raised comments have addressed accordingly.

Reviewer 1

Comment

Answer

Location

In material section, please mention the grade of HPMC.

Thank you for your comment. The grade of HPMC used in this study was HPMC(K 100 M)

In section 2.1, highlighted by track change function.

What was the rational of using the specific grade of HPMC?

Thank you for your comment.

HPMC (K 100 M) was selected for our patch formulations because our intention was to achieve slow and control release of drug on weekly basis. The HPMC grades such as; (K 4 M and K 15 M) don’t provide the resultant controlled release for the desire time, so high viscosity grade HPMC (K 100 M) was selected.

The grade of HPMC (K 100 M) and its rational of using in TDDS patches have been added to the section 1 and highlighted by track change function.

In skin irritation study, please mention the relative humidity.

Thank you for your comment.

The temperature was maintained at 25 ± 2 â—¦C with relative humidity controlled at 50 ± 10% for all rabbits.

The temperature and relative humidity is incorporated to the section 2.6 and highlighted by track change function.

For in vitro drug release study, please mention properties of Tuffryn membrane, such as thickness of membrane, pore size or cut-off MW etc.

Thank you for your comment.

Tuffryn membrane used for in vitro study was having diameter, 2.5 mm and pore size of 0.45 µm. 

The diameter and pore size of Tuffryn membrane is added in the section 2.7 and highlighted by track change function.

Why Power Law model for spherical shape was considered for transdermal patch? The 'n' value should corresponds to flat surface (film) and the subsequent discussion should be based on it.

Thank you for your comment.

The release data were fitted into different kinetic models to investigate the kinetic behaviour as well as release mechanism of drug from the patch formulations. However, the release data gave optimum R2 value with power law equation and it is also advantageous to describe the mechanism of drug release.

Linear equation was showed by all formulated patches. Power law equation was better suited to our data. Anomalous, non ̶ Fickian release mechanism was observed for MTX patches. Added to the section 3.6 and highlighted by track change function.

Reviewer 2 Report

This manuscript is devoted to the development of  topical patch  with Methotrexate substance that was chosen for its efficacy as well as evaluation of its better penetration and augmented pharmacokinetics. Topical patch formulations containing methotrexate by employing various concentration of HPMC and Eudragit S-100 polymers as well as optimum concentrations of various penetration enhancers were studied. All transdermal patches with and without penetration enhancers showed good physicochemical properties. The study showed that in vitro drug release directly proportional to the use of polymer type.

I would suggest to add In the Introduction section, several critical reviews should be added in the revised form such as:  (2021). Stimuli-Responsive Polymers for Transdermal, Transmucosal and Ocular Drug Delivery. Pharmaceutics, 13(12), 2050.

Dabholkar, N., Gorantla, S., Waghule, T., Rapalli, V. K., Kothuru, A., Goel, S., & Singhvi, G. (2021). Biodegradable microneedles fabricated with carbohydrates and proteins: Revolutionary approach for transdermal drug delivery. International Journal of Biological Macromolecules, 170, 602-621

Lines 420-440. Percentage Moisture Content I would suggest to mention related work of hydrophilic scaffold

"Water uptake as a crucial factor on the properties of cryogels of gelatine cross-linked by dextran dialdehyde." Gels 7.4 (2021): 159.

Lines 250-251 The drug release mechanisms of formulated transdermal patches of methotrexate were evaluated with the use of following kinetic models [15, 16]. Please provide brief description and equations

Please provide the reference to equations 5-10 from the original source, persons who discover and created it. 

Lines 434-435 The formulated patches exhibited drug contents in the range of 92.57 ± 3.22 to 98.37 ± 0.56. units should be included

Line 448 Folding endurance of formulated patches showed 61 ± 3.14 to 74 ± 2.32. units should be included.

Line  466 Table 4. Stability studies of MTX patches.

How did authors keep constant RH 60 and 75%?

 Line  495 Hence, F5 showed maximum amount of drug drug Please correct it

Lines 644-645 The moisture content values for the formulated patches ranged in between 11.18 ± 1.54 to 12.77 ± 1.12 please include units

Lines 651-652 The uniform drug distribution in the formulated transdermal patches results in achieving sustained or controlled drug delivery. The drug content of prepared patches were from 95.66 ± 2.43 to 98.87 ± 2.21 please include units

enhancers (eucalyptus oil, N-methyl-2-pyyrolidone and Oleic acid)  Please  correct typo

Author Response

Reply to the comments of Reviewer 2.

Greetings of the day!

First of all I am really indebted to you for your kind consideration of our research article. The valuable suggestions from your kind self-regarding our research article reflected your thorough grounding and expertise in the specialized field for which I am much grateful to you.

Reviewer 2

Comment

Answer

Location

I would suggest to add In the Introduction section, several critical reviews should be added in the revised form such as:  (2021). Stimuli-Responsive Polymers for Transdermal, Transmucosal and Ocular Drug Delivery. Pharmaceutics13(12), 2050.

Thank you for your comment.

The critical reviews have been added in the introduction portion of the paper.

The critical reviews have been added to the section 1(Introduction) and highlighted by track change function. The suggested article has been cited as [7].

Dabholkar, N., Gorantla, S., Waghule, T., Rapalli, V. K., Kothuru, A., Goel, S., & Singhvi, G. (2021). Biodegradable microneedles fabricated with carbohydrates and proteins: Revolutionary approach for transdermal drug delivery. International Journal of Biological Macromolecules170, 602-621.

Thank you for your comment.

Some portion from this paper has been added to the introduction portion.

Some introduction portion is added to the section-1(Introduction) and highlighted by track change function. The suggested article has been cited as [2].

Lines 420-440. Percentage Moisture Content I would suggest to mention related work of hydrophilic scaffold.

Thank you for your comment.

Since HPMC is hydrophilic in nature and presence of maximum concentration of HPMC results in greater amount of moisture absorption. The statement has been corrected accordingly.

The said statement has been corrected as directed by your kind self in section 3.2 and highlighted by track change function.

"Water uptake as a crucial factor on the properties of cryogels of gelatine cross-linked by dextran dialdehyde." Gels 7.4 (2021): 159.

Thank you for your comment.

The formulations containing maximum amount of HPMC polymers shows higher value for water uptake. The reason might be due to the presence of maximum amount of hydrophilic polymer (HPMC).

The suggested article has been cited as [24].

Please provide the reference to equations 5-10 from the original source, persons who discover and created it. 

Thank you for your comment.

The references from original source have been added to the said equations (5-10) with name of discoverer.

The references from original source have been added to the said equations (5-10) with name of discoverer at section 2.8.1, 2.8.2, 2.8.2, 2.8.4 and 2.8.5 and highlighted with track change function .

Lines 434-435 The formulated patches exhibited drug contents in the range of 92.57 ± 3.22 to 98.37 ± 0.56. units should be included.

Thank you for your comment.

The unit for drug content estimation is %. So, the % unit is added to the values accordingly.

The unit for drug content estimation is %. So, the % unit is added to the values accordingly in the section 3.2 and highlighted with track change function.

Line 448 Folding endurance of formulated patches showed 61 ± 3.14 to 74 ± 2.32. units should be included.

Thank you for your comment.

The number of folding of patch until crack or break appears is the folding value of patch.

The number of folding of patch until crack or break appears is the folding value of patch. The unit of folding endurance has been added in section 2.4.4 and highlighted with track change function.

Line  466 Table 4. Stability studies of MTX patches.

Thank you for your comment.

Corrected as directed.

Corrected as directed in the section 3.3 and highlighted with track change function.

How did authors keep constant RH 60 and 75%?

Thank you for your comment.

The procedure for maintaining RH value 60±5% and 75±5% has been added to the methodology portion.

The procedure for maintaining RH value 60±5% and 75±5% has been added to the methodology portion section 2.5 and highlighted with track change function.

Line  495 Hence, F5 showed maximum amount of drug drug Please correct it

Thank you for your comment.

The duplication of word drug has been removed.

The duplication of word drug has been removed in the section 3.5 and highlighted with track change function.

Lines 644-645 The moisture content values for the formulated patches ranged in between 11.18 ± 1.54 to 12.77 ± 1.12 please include units

Thank you for your comment.

The unit for moisture content estimation is %. So, the % unit is added to the values accordingly.

The unit for moisture content estimation is %. So, the % unit is added to the values accordingly in the section 3.11 and highlighted with track change function.

Lines 651-652 The uniform drug distribution in the formulated transdermal patches results in achieving sustained or controlled drug delivery. The drug content of prepared patches were from 95.66 ± 2.43 to 98.87 ± 2.21 please include units

Thank you for your comment.

The unit for Drug content estimation is %. So, the % unit is added to the values accordingly.

The unit for Drug content estimation is %. So, the % unit is added to the values accordingly in the section 3.11 and highlighted with track change function.

enhancers (eucalyptus oil, N-methyl-2-pyyrolidone and Oleic acid)  Please  correct typo

Thank you for your comment.

Penetration enhancers (Eucalyptus oil, N-methyl-2-pyrrolidone and Oleic acid) have been corrected throughout manuscript.

The Penetration enhancers (Eucalyptus oil, N-methyl-2-pyrrolidone and Oleic acid) have been corrected throughout manuscript in Sections: 2.1, 3.10, 3.12, 3.13, 3.14, and 3.15 accordingly.

Highlighted with track change function.

Round 2

Reviewer 1 Report

The manuscript can now be accepted in its present form.

Reviewer 2 Report

Lines 184-185 Following equation best demonstrate folding endurance.
F=d (1)

where F is folding endurance and d is number of folds required to break or crack the patch

Please check the equation.